# Insight into the Effect of Ice Addition on the Gel Properties of *Nemipterus virgatus* Surimi Gel Combined with Water Migration

**DOI:** 10.3390/foods10081815

**Published:** 2021-08-05

**Authors:** Haiqiang Chen, Yiqian Zou, Aimei Zhou, Jie Xiao, Soottawat Benjakul

**Affiliations:** 1College of Food Science, South China Agricultural University, Guangzhou 510642, China; chq888120615@163.com (H.C.); zouyiqian29@163.com (Y.Z.); xiaojieacademic@163.com (J.X.); 2Guangdong Provincial Key Laboratory of Nutraceuticals and Functional Foods, Guangzhou 510642, China; 3Department of Food Technology, Faculty of Agro-Industry, Prince of Songkla University, Hat Yai 90112, Songkhla, Thailand; soottawat.b@psu.ac.th

**Keywords:** the amount of ice added, water migration, gel properties, *Nemipterus virgatus* surimi, two-stage heat treatment

## Abstract

The effect of the amount of ice added (20–60%) on the gel properties and water migration of *Nemipterus virgatus* surimi gel obtained with two-stage heat treatment was studied. The gel strength and water-holding capability (WHC) of the surimi gel with 30% ice added were significantly higher than those of other treatment groups (*p* < 0.05). The addition of 30% ice was conducive to the increase of protein β-sheet proportion during heat treatment, exposing more reactive sulfhydryl groups. These promoted the combination of protein-protein through disulfide bonds and hydrophobic-hydrophobic interactions, forming an ordered three-dimensional gel network structure. Meanwhile, the increase in hydrogen bonds promoted the protein-water interaction. Low-field nuclear magnetic resonance analysis showed that more bound water was locked in the gel system, reducing the migration of immobile water to free water and finally showing better gel properties. When the amount of ice added was insufficient (20%), the gel structure lacked the support of immobile water, resulting in deterioration of gel strength. However, excessive addition of ice (>30%) was not conducive to the combination of protein-protein and protein-water, forming a large and rough gel structure, resulting in the migration of immobile water to free water and ultimately exhibited weak gel properties.

## 1. Introduction

Surimi is an intermediate product of many seafood products, which is a refined and stable fish myofibril protein after multiple processes. The myofibrillar protein of surimi unfolds, stretches, and aggregates during heating, forming a three-dimensional network structure and trapping water molecules, thereby increasing the gel properties of surimi products [1]. Existing research generally believes that the gel properties of surimi gel are determined by the gel network structure formed by protein denaturation and aggregation [2,3]. However, water molecules bound to the protein provide a hydration environment for myofibrillar protein and stabilize the three-dimensional structure of the gel [4]. In addition, the water molecules locked by the gel network structure may play a supporting role and affect the gel properties of the surimi gel. Therefore, it is speculated that the state and distribution of water molecules in the gel system also play a key role in the improvement of surimi gel properties. At present, low-field nuclear magnetic resonance (LF-NMR) is a powerful technique to characterize the state, distribution, and mobility of water molecules in biological macromolecule systems [5]. It is a non-destructiveness detection and quantitative visualization technology, which has been applied to the study of water molecules in various food materials [6]. According to the relaxation time of hydrogen protons (T_2_), the state of water molecules in the gel system is divided into bound water, immobile water, and free water [7]. The water migration in the gel system affects the state and distribution of water molecules.

Under the same heat treatment conditions, the different moisture contents may also affect the water migration of surimi gel. However, since the temperature of surimi paste is required to be lower than 10 °C during the preparation process to maintain the gelling ability of fish protein and prevent its denaturation [8], and most surimi product processing factories do not use chopping equipment with cooling function. Therefore, frozen surimi is usually thawed to a semi-thawed state, and ice is used instead of water to offset the heat generated during the processing of surimi products. Up to now, the mechanism of the amount of ice added to the gel properties of surimi is still unclear. Although some studies have reported the influence of different moisture content on the gel strength of surimi [9,10,11], the gel-forming method in these studies does not use a two-stage heat treatment (40 °C/30 min, 90 °C/20 min). Nevertheless, the two-stage heat treatment method is the most commonly used method for surimi gel processing and has been used in the surimi industry for decades [12]. Therefore, to clarify, the effect of the amount of ice added on the properties of surimi gel under the two-stage heat treatment has an important role in promoting the development of the surimi industry.

*Nemipterus virgatus* (*N. virgatus*) is a major low-value fish in Southeast Asian countries. it is commonly used in surimi due to its high production and high protein content [13]. This study explored the effect of different amounts of ice added (20–60%) on the gel properties of *N. virgatus* surimi gel under the same heat treatment conditions (two-stage heat treatment), and combined with water migration to clarify the effect of ice addition on the gel properties of surimi gel. Thus, the present study aims to confirm the influence of water migration on the gel properties of surimi, clarify the mechanism of the effect of ice addition on the gel properties of two-stage heat-treated surimi, and provide a reference for the processing of surimi products of other fish species.

## 2. Materials and Methods

### 2.1. Materials and Reagents

*N. virgatus* surimi (grade AAA) was purchased from Yangjiang Xinjiang Aquatic Products Refrigeration Factory Co., Ltd. (Yangjiang, Guangdong, China) and stored at −20 °C. The moisture content of the surimi was 74.5 ± 0.1%. The crushed ice was prepared using an ice maker (Scientz XB-100, Ningbo, China). Other reagents were of analytical grade, purchased from Guangzhou Chemical Reagent Factory (Guangdong, China).

### 2.2. Sample Preparation

Frozen *N. virgatus* surimi with vacuum packaging bag was thawed in water for 30 min and then cut into small pieces of about 1 cm. A mixer (Shanghai Specimen Model Co., Shanghai, China) was then used to chop these small squares for 1 min followed by the addition of 2.5% salt (*w*/*w*) and different amounts of ice (20%, 30%, 40%, 50%, and 60%, *w*/*w*). The final moisture content of the surimi was 76.9%, 78.6%, 79.9%, 81.4%, and 82.6%, respectively. The determination of moisture content was based on the AOAC procedure (985.29) [14]. The surimi was further chopped for 4 min to obtain a uniform paste, and then the surimi pastes were stuffed into polyvinylidene chloride casings (30 mm i.d, length 200 mm), and the ends were tightly sealed with cotton rope. Two-stage heat treatment (40 °C/30 min and 90 °C/20 min) was used to prepare surimi gels. The prepared surimi gels were placed in an ice-water mixture to cool for 1 h and stored at 4 °C for 12 h before being analyzed.

### 2.3. Breaking Force, Deformation, and Gel Strength

The gel strength of the surimi gel was calculated by multiplying breaking force and deformation. The mechanical properties, including breaking force (g) and deformation (mm), of samples (20 mm high cylinders) were measured using a CT-3 texture analyzer (Brookfield Engineering Laboratories, Inc.Brookfield, WI, USA) with a P/5 flat-surface cylindrical probe (5 mm in diameter) at room temperature (25 °C). The trigger type was set to auto (force). The setting parameters were as follows: sensing force, 4.0 g; pre-compression speed, 2.0 mm/s; testing speed, 1.0 mm/s; pressing distance, 15.0 mm; and a recovery speed of 1.0 mm/s. Each group of surimi gel samples was analyzed seven times.

### 2.4. Water-Holding Capacity (WHC) and Whiteness

The WHC of the surimi gel samples was evaluated based on the method proposed by Kocher & Foegeding [15] with some modifications. The surimi gel sample (0.5 cm × 0.5 cm × 2 cm square column) was accurately weighed (GG), transferred to a two-layer filter paper, and placed in a 5 mL centrifuge tube (inner tube) with four holes at the bottom. Then the inner tube (containing the gel sample) was placed in a centrifuge tube (15 mL, outer tube), and centrifuged at 10,000× *g* at 4 °C for 10 min (Eppendrof, centrifuge 5804 R, Hamburg, Germany). After removing the filter paper, the weight of the surimi gel sample (GG’) was recorded. Each group of surimi gel samples was tested, in parallel, six times. WHC (%) was calculated as follows:(1)$$\mathrm{WHC}\left(\%\right)=\left(1-\frac{\mathrm{GG}-{\mathrm{GG}}^{\prime }}{\mathrm{GG}}\right)\times 100$$

The lightness (*L**), red-green value (*a**), and yellow-bule value (*b**) of the gel samples were determined using a colorimeter (CR-410, Konica Minolta Camera, Co, Japan). The whiteness was calculated as follows [4]:(2)$$\mathrm{Whiteness}=100-{\left[{\left(100-L^{*}\right)}^{2}+a^{*2}+b^{*2}\right]}^{1/2}$$

### 2.5. Sodium Dodecyl Sulfate-Polyacrylamide Gel Electrophoresis (SDS-PAGE)

Protein patterns of surimi gels were examined by SDS-PAGE according to the method of Kudre, Benjakul, and Kishimura [16] with some modifications. 27 mL sodium dodecyl sulfate (SDS, 5% *w*/*v*) was added to the mashed surimi gel sample (3 g) and homogenized for 2 min at 12,000 rpm (T18, IKA, Wertheim, Germany). The homogenized mixture was placed in a water bath (85 °C), heated for 1 h, cooled, and centrifuged at 8000× *g* for 20 min (Eppendrof, centrifuge 5804 R, Hamburg, Germany). The protein concentration of the supernatant was determined by the biuret method. After the protein concentration of the supernatant was diluted to 3 mg/mL, the supernatant and 5 × SDS-PAGE buffer (10% SDS, 7.5% dithiothreitol, 50% glycerol, 0.5% bromophenol blue, 0.25 M Tris-HCl buffer, pH 6.8) were mixed at a ratio of 1:4 (*v*/*v*) and heated in boiling water for 5 min. The mixture (5 μL) was loaded onto the poly-acrylamide gel (containing 4% stacking gel and 4–20% gradient running gel), and the gel was run at a constant voltage of 180 V. After separation, the electrophoresis gel was stained with 0.10 g/100 mL Coomassie Blue R 250 in 50 mL/100 mL methanol and 7.5 mL/100 mL acetic acid for 1 h. The gel was then decolorized with a decolorizing solution (glacial acetic acid, distilled water, and methanol at the ratio of 8:1:1) and observed in a Bio-Rad imaging system (ChemiDoc XRS+, Hercules, CA, USA). ImageJ software was used for density analysis of protein bands.

### 2.6. Fourier-Transform Infrared (FT-IR) Spectroscopy

The surimi gel sample was dried in a vacuum freeze dryer (Christ, ALPHA2-4 LDplus, Harz, Germany), and the dried sample was ground into a powder with a mortar. The gel sample powder (1–2 mg) and KBr (20–30 mg) were mixed and ground into a fine powder, which was then made into a thin sheet by a tablet machine (Jingsheng Instrument, 769YP-24B, Shanghai, China). The FT-IR spectroscopy (Bruker, Vertex 70, Karlsruhe, Germany) was used to record the spectrum of the sheet in the range of 400–4000 cm^−1^, and the sample sheet without KBr was used as the blank with air as background. EZ Omnic 7.3 (Thermo Electron Corporation, Madison, WI, USA) was used to export the data of amide I band (1700–1600 cm^−1^), and the protein secondary structure was analyzed using PeakFit software 4.12 (SeaSolve, Framigham, MA, USA).

### 2.7. Chemical Interactions

The chemical interaction determination of the surimi gel sample was carried out based on the method of Gómez-Guillén, Borderías, and Montero [17] with some modifications. Different denaturing solutions (10 mL) were added to the mashed gel sample (2 g), including SA (0.05 mol/L NaCl), SB (0.6 mol/L NaCl), SC (0.6 mol/L NaCl and 1.5 mol/L urea), SD (0.6 mol/L NaCl and 8 mol/L urea), and SE (0.6 mol/L NaCl, 8 mol/L urea and 0.5 mol/L β-mercaptoethanol). The mixtures were homogenized at 12,000 rpm for 2 min (T18, IKA, Germany), and incubated at 4 °C with shaking for 1 h, followed by centrifugation at 10,000× *g* for 15 min (Eppendrof, centrifuge 5804 R, Hamburg, Germany). The Bradford method was used to determine the protein concentration of the supernatant after centrifugation. The calculation formula of chemical interactions was as follows:Nonspecific bonds = c(SA)Ionic bonds = c(SB-SA)Hydrogen bonds = c(SC-SB)Hydrophobic interactions = c(SD-SC)Disulfide bonds = c(SE-SD)(3)
where the unit of chemical interactions was expressed as g protein/L solutions; c(SA) was the protein concentration of the supernatant of the sample containing the SA solution; c(SB-SA) was the difference in the protein concentrations of the centrifugal supernatants of the sample containing SB and SA solutions. In addition, c(SC-SB), c(SD-SC), and c(SE-SD) were also obtained in the same manner.

### 2.8. Reactive Sulfhydryl (R-SH) and Surface Hydrophobicity

The mixture of surimi gel sample (3 g) and phosphate buffer solution (27 mL, 100 mM, pH 7.6) was homogenized at 12,000 rpm for 2 min (T18, IKA, Germany), and then centrifuged at 10,000× *g* for 15 min (Eppendrof, centrifuge 5804 R, Hamburg, Germany). The supernatant was collected to determine the R-SH group and surface hydrophobicity.

The R-SH group was measured according to the method of Ellman [18] with a slight modification. Briefly, 0.6 mL of the centrifuged supernatant was added to 2.4 mL of Tris-glycine buffer (0.1 M Tris, 0.1 M glycine, 4 mM ethylenediaminetetraacetic acid, pH 8.0) and mixed thoroughly. Then 0.02 mL of Ellman’s reagent (4 mg/mL 5,5′-Dithiobis-(2-nitrobenzoic acid) solution in Tris-glycine buffer) was added to the mixture and incubated at 40 °C for 15 min. The absorbance of the mixed solution at 412 nm was measured by Enspire microplate reader (Perkin Elmer, Inc., Baesweiler, Germany), and the R-SH group was calculated using the following formula:(4)$$C_{0}=\frac{A}{\varepsilon \times C}\times D$$
where *C*_0_ was the molarity of R-SH, mol/g; *A* was the absorbance at 412 nm of the mixed solution; *ε* was the molar extinction coefficient, 13,600 L/mol·cm; *c* was the concentration of protein (mg/mL); *D* was the dilution ratio.

The surface hydrophobicity (S_0_) of the surimi gel sample was measured according to the method of Zhang et al. [19] with 1-anilinonaphthalene-8-sulfonic acid (ANS) as a hydrophobic fluorescent probe. The centrifugal supernatant was diluted into a series of solutions at different protein concentrations (0.1 to 0.5 mg/mL), and 25 µL of ANS reagent (8 mM in 0.1 M phosphate buffer, pH 6.0) was added to the sample solutions with different protein concentrations. After mixing, the mixtures were incubated in the dark for 20 min (25 °C). The fluorescence intensity of the mixture solutions was measured using a RF-6000 fluorescence spectrophotometer (Shimadzu, Kyoto, Japan) at an emission wavelength of 485 nm and an excitation wavelength of 374 nm. Surface hydrophobicity (S_0_-ANS) was expressed as the initial slope of fluorescence intensity to protein concentrations.

### 2.9. Myofibrillar Protein Content

The determination method of the myofibrillar protein content of surimi for different amounts of ice was carried out according to a previous report [20]. All steps of myofibril protein extraction were performed at a temperature below 4 °C, and the speeds of homogenization and centrifugation were 10,000 rpm and 12,000 rpm, respectively. A surimi sample (2 g) was added with distilled water (15 mL, 4 °C) and homogenized for 2 min (T18, IKA, Germany), followed by centrifugation at 4 °C for 10 min (Eppendrof, centrifuge 5804 R, Hamburg, Germany). NaCl solution (15 mL, 0.3%, *w*/*v*) was added to the centrifuged pellet, homogenized for 2 min, and then centrifuged at 4 °C for 10 min. Tris-maleate buffer (30 mL, 20 mM, dissolved in 0.6 M NaCl) was added to the collected centrifuged pellet, homogenized for 10 min, then incubated at 4 °C for 1 h, and centrifuged for 10 min. Distilled water (32 mL, 4 °C) was then added to the supernatant (8 mL) and centrifuged for 10 min. The collected precipitate (myofibrillar protein) was dissolved in 0.6M NaCl (pH 7.0), centrifuged for 5 min to remove the insoluble matter, and the myofibrillar protein content of the supernatant was measured by the biuret method.

### 2.10. Water Mobility and Distribution of Surimi Gel

The water mobility and distribution of surimi gel were analyzed according to the method of Wang, Zhang, Bhandari, and Gao [21], using an LF-NMR analyzer (Niumang Analytical Instrument Corporation, Shanghai, China) with a magnetic field strength of 0.5 T. The surimi gel was cut into a cylinder with a height of 30 mm (30 mm i.d) and placed in a test tube (50 mm i.d), followed by spin-spin relaxation time (T_2_) analysis at 32 °C.

### 2.11. Scanning Electron Microscopy (SEM)

The surimi gel was made into 3 mm × 3 mm × 3 mm cubes, fixed with 2.5% (*v*/*v*) glutaraldehyde for 12 h, and then immersed in 0.1 M phosphate buffer (pH 7.2) for 10 min. The fixed gel sample was dehydrated in the ethanol solutions of different concentrations (30%, 50%, 70%, 80%, 90%, and 100%) for 10 min each time. The dehydrated gel sample was dried at the critical point with CO_2_ as a transition fluid and fixed on a bronze stub for gold spraying. Finally, an SEM (LEO 1530 VP, Carl Zeiss, Jena, Germany) was used to visually observe and image the sample.

### 2.12. Statistical Analysis

Origin-pro 9.1 and Photoshop CS5 software were used to draw and assemble graphics, respectively, and SPSS 19.0 software was used for one-way Analysis of Variance (ANOVA) between data to obtain statistical differences between data. The expression format of the experimental data was mean ± standard deviation (SD).

## 3. Results and Discussion

### 3.1. Effect of the Amount of Ice Added on the Breaking Force, Deformation, and Gel Strength of Surimi Gel

As shown in Figure 1a, the breaking force and deformation of the surimi gel added with 30% ice were significantly higher than those of other treatment groups (*p* < 0.05). When the amount of ice added exceeded 30%, the breaking force of the surimi gel decreased significantly with the increase of ice addition level (*p* < 0.05), which was inconsistent with the results of previous studies [9]. Park et al. [9] found that the breaking force of *N. virgatus* surimi gel gradually decreased with the increase of moisture content. Different surimi gel preparation methods (40 °C/30 min, 90 °C/20 min vs. 90 °C/15 min) might account for this difference. The preparation method of surimi gel in this study adopted a two-stage heat treatment. The setting process (40 °C/30 min) in two-stage heat treatment was conducive to protein-protein cross-linking catalyzed by the endogenous transglutaminase (TGase), which played a vital role in the formation of the gel [12]. The deformation of the surimi gel with 20% ice added was only 0.86 ± 0.03 mm, while the deformation of the gels added with a higher amount of ice (≥30%) increased significantly to 1.02–1.22 mm. This might be attributed to the lack of sufficient water molecule support for the 20% ice-added gel sample, resulting in week gel elasticity and easy puncture. Although the deformation of the surimi gel with 60% ice added was significantly higher than that of the gel with 50% ice added (*p* < 0.05), its breaking force was significantly reduced (*p* < 0.05). When the amount of ice added was 30%, the gel strength (709.30 ± 40.32 g·cm) of surimi gel was significantly higher than that of other treatment groups (*p* < 0.05) (Figure 1b), and the gel strength was only 416.62 ± 29.64 g·cm and 302.78 ± 17.24 g·cm when the ice was added with 20% and 60% respectively, which was 58.75% and 42.69% of that of the sample with 30% ice added.

Previous studies believed that myofibrillar protein unfolded and aggregated during heating, forming a three-dimensional network structure, entrapping water molecules, and forming a surimi gel with good gel properties [1]. When the amount of ice added was too low (20%), there were not enough water molecules locked in the gel structure, and the gel strength of the surimi gel became weak. Therefore, water molecules might play a supporting role in the gel system, and an appropriate amount of ice added was more conducive to improving the mechanical properties of the gel structure. When the amount of ice added increased, the moisture content of surimi increased, and the concentration of myofibrillar protein was too low to maintain a sufficiently strong three-dimensional network structure, which was not advantageous to improving the gel strength of gel [9]. In addition, the thermal diffusivity of surimi paste increased significantly with the increase of moisture content [22], while the increase in thermal diffusivity of surimi paste did not facilitate the complete expansion of the protein and the exposure of the hydrophobic domain, which further weakened the protein-protein interaction [23].

### 3.2. Effect of the Amount of Ice Added on the WHC and Whiteness of Surimi Gel

The WHC of the surimi gel with 30% ice added was significantly higher than the other treatment groups as depicted in Figure 1c (*p* < 0.05), which was consistent with the results of the gel strength. When the amount of ice added exceeded 30%, the WHC of surimi gel decreased significantly with the increase of ice amount (*p* < 0.05). The moisture content of surimi increased with the increase in the amount of ice added, which might lead to a decrease in the concentration of myofibril protein or an increase in the thermal diffusion rate of surimi gel. These were not in favor of the interaction between proteins, thereby affecting the formation of gel structure [9,23].

The whiteness of surimi gel increased with the increase in the amount of ice added (Figure 1d), which was consistent with the results of previous studies [11]. The whiteness of surimi gel and the change of gel strength were inconsistent, indicating that the correlation between gel strength and whiteness was low under different ice additions. The lightness (*L**) of surimi gel increased with the increase of ice addition, but the yellow-bule value (*b**) showed the opposite trend (Table 1), which was consistent with the results of previous studies [24]. These indicated that the higher the amount of ice added, the lighter and the less yellow the surimi gel. In summary, the whiteness of surimi gel was mainly affected by the *L** and *b** values under different ice additions.

### 3.3. Effect of the Amount of Ice Added on SDS-PAGE of Surimi Gel

As shown in Table 1 and Figure 2, the density of myosin heavy chain (MHC) bands of surimi gel increased significantly with the increase of the ice addition level (*p* < 0.05). The MHC band density of the 20% and 30% ice-added surimi gels was only half of that of the surimi gels added with a higher amount of ice. On the other hand, the actin band density of the 30% ice-added surimi gel was significantly higher than that of the other treatment groups (*p* < 0.05), which was consistent with the change of gel strength. Myosin and actin were the main proteins that form surimi gel [25], and the decrease of MHC band density was accompanied by the improvement of mechanical properties [26]. In the setting stage (40 °C/30 min) of heat treatment, the combination of myosin and actin promoted the formation of the three-dimensional network structure of the gel. The decrease in MHC band density and the increase in actin band density might contribute to the improvement of the gel strength of the surimi gel with 30% ice added. As the amount of ice added increased, the density of the top bands of spacer gel gradually increased, which indicated that the formation of large molecular weight proteins in the gel gradually increased [27]. The thermal diffusion of the surimi increased with the increasing moisture content [22], which led to the rapid aggregation of proteins to form macroaggregates.

### 3.4. Effect of the Amount of Ice Added on the Secondary Structures of Surimi Gel Protein

The protein of the surimi gel with 30% ice added had the highest β-sheet proportion, the lowest α-helix, and random coils proportion as demonstrated in Table 2, which were significantly different from other treatment groups (*p* < 0.05). These indicated that the increase in the β-sheet proportion of the surimi gel protein with an appropriate amount of ice (30%) was contributed by the unfolding of α-helix and the orderliness of the random coils. The β-sheet proportion of surimi gel protein decreased with the increase of ice addition, which might be caused by the increase of the thermal diffusivity of surimi paste [22,23]. Compared with α-helix, the β-sheet had a looser structure and exposed more side chains, which was conducive to protein-protein interaction during heat treatment [28].

### 3.5. Effect of the Amount of Ice Added on the Chemical Interactions of Surimi Gel

As illustrated in Figure 3a, the hydrophobic interactions were the highest chemical interaction force in surimi gel with different ice additions. Though there was no significant difference in hydrophobic interactions between the surimi gels with 30% and 40% ice added (*p* > 0.05), but the hydrophobic interactions of the surimi gel with 30% ice were significantly higher than other treatment groups (*p* < 0.05). In addition, the ionic bonds, hydrogen bonds, and disulfide bonds of the 30% ice-added surimi gel were also significantly higher than those of other treatment groups (*p* < 0.05). Hydrophobic-hydrophobic interactions and disulfide bonds played a crucial role in protein-protein interactions. They were the main forces for stabilizing the heat-induced gel structure [29]. Disulfide bonds, a kind of stable covalent bond, were formed due to the unfolding of protein and the exposure of buried active sulfhydryl groups [30]. Hydrogen bonds and ionic bonds (electrostatic interactions) were an important force in the interaction between biological macromolecules and water molecules and had an important contribution to the gel system locking the water molecules [31]. With the increase of the amount of ice added (>30%), the chemical interaction forces in the surimi gel gradually weakened, which might be due to the increase in the thermal diffusivity of the surimi paste [22]. Under the influence of the addition level of ice, the changing trend of the four chemical interaction forces was consistent with the results of gel strength, and the excellent mechanical properties and the ability to capture water molecules of surimi gel were important factors for improving the gel strength. Therefore, it was speculated that the increase in hydrophobic-hydrophobic interactions and disulfide bonds resulted in outstanding mechanical properties of the 30% ice added surimi gel, and the enhancement of hydrogen bonds and electrostatic interactions contributed to the excellent water molecule capture ability of surimi gel.

### 3.6. Effect of the Amount of Ice Added on the Content of R-SH Group and Surface Hydrophobicity of Surimi Gel

The effect of ice addition on the content of R-SH group of surimi gel was shown in Figure 3b. The level of R-SH group of the 30% ice-added surimi gel (8.38 ± 0.06 mmol/10^6^ g protein) was significantly higher than that of the surimi gel added with 20% ice (7.71 ± 0.05 mmol/10^6^ g protein) (*p* < 0.05). When the addition level of ice exceeded 30%, the content of R-SH group decreased significantly with the increase of ice amount (*p* < 0.05). The S_0_-ANS of the surimi gel with 30% ice added was 1629.87 ± 27.78, which was 96.11% higher than that of the surimi gel with 20% ice added (*p* < 0.05) (Figure 3c). However, the S_0_-ANS value of the 40% ice-added surimi gel was 24.19% lower than that of the 30% ice-added surimi gel (*p* < 0.05). When the amount of ice added was higher than 40%, the S_0_-ANS of the surimi gel gradually decreased, but they were still significantly higher than that of the surimi with 20% ice added (*p* < 0.05).

During the heat treatment, the unfolding of the protein facilitated the exposure of the amino acid side chains hidden in the hydrophobic region [3], thereby exhibiting a higher content of R-SH group. It was well known that the exposure of R-SH group played a key role in the formation of disulfide bonds [32]. In addition, when the hydrophobic core of the protein was decomposed, more 8-anilino-1-naphthalene sulphonic acid (ANS) bound to the previously masked hydrophobic region [33], leading to higher S_0_-ANS. These indicated that the surimi gel with 30% ice added had more hydrophobic-hydrophobic interactions during the heat treatment stage. Numerous hydrophobic-hydrophobic interactions promoted the stability of the protein network structure and improved the texture properties of the surimi gel [34].

### 3.7. Effect of the Amount of Ice Added on the Myofibrillar Protein Content of Surimi

As depicted in Figure 3d, the myofibril protein concentration of the surimi with 20% ice added was 43.38 ± 4.01 mg/g, which was significantly higher than that of the surimi with 30% ice (38.08 ± 2.72 mg/g). When the amount of ice added was above 30%, the myofibrillar protein content of surimi gradually decreased with the increase of ice added, but the difference in myofibrillar protein concentration in these samples was not significant (*p* > 0.05). The decrease in the concentration of myofibrils reduced the possibility of their formation of a three-dimensional network [9], resulting in the deterioration of the gel properties of surimi. However, the gel strength of the 20% ice-added surimi gel was significantly lower than that of the surimi gel added with 30% ice (*p* < 0.05) (Figure 1b), though its concentration of myofibrillar protein was significantly higher than that of the latter (*p* < 0.05). This might be the reason that the amount of water molecules also played a key role in the gel strength of surimi gel. Although the concentration of myofibrillar protein in surimi was high, the lack of sufficient water molecule support rendered weak gel properties. The content of myofibrillar protein did not change significantly with the increase of ice addition (≥30%), but the gel strength of the surimi gel with 30% ice added was significantly higher than that of surimi gels with a higher amount of ice added. This demonstrated that the main driving force which caused the decrease in the gel strength of surimi gel with the increase of ice addition might be the increase in thermal diffusivity of surimi paste rather than the decrease in myofibrillar protein concentration.

### 3.8. Effect of the Amount of Ice Added on the Water Mobility and Distribution of Surimi Gel

The spin-spin relaxation time (T_2_) of hydrogen protons is used to estimate the mobility and distribution of water molecules in the gel system. The transverse relaxation times (T_21_, T_22_, and T_23_) of hydrogen protons represent the water molecules of bound water, immobile water, and free water, respectively, and their corresponding peak area populations (P_21_, P_22_, and P_23_) reveal the changes in the distribution of the three kinds of water molecules [7,35]. The shorter the relaxation time (T_2_), the stronger the binding force of water molecules. As the amount of ice added increased, the bound water (T_21_) peak of surimi gel gradually changed from a single peak to two folding peaks as depicted in Figure 4 and Table 3, indicating that the binding force on the bound water gradually weakened. Compared with the 30% ice-added surimi gel, the surimi gel with 20% ice added had a significant decrease in T_22_ and T_23_ (*p* < 0.05), illustrating that the bound water and immobile water of the surimi gel with 20% ice were bound by a stronger binding force than the surimi gel with 30% ice. However, the P_22_ of the surimi gel with 30% ice added was significantly higher than that of the surimi gel with 20% ice (*p* < 0.05), which was consistent with the gel strength results, demonstrating that the increase in the relative content of the immobile water resulted in better gel strength of the surimi gel with 30% ice than that of the surimi gel with 20% ice. Therefore, it could be inferred that water molecules (immobile water) might play a supporting role in the gel network structure and also determined the improvement of the gel strength.

### 3.9. Effect of the Amount of Ice Added on The Microstructure of Surimi Gel

As shown in Figure 5, the pores of the surimi gel with 20% ice added were mostly flat and irregular, which might be caused by the lack of immobile water support during the gel formation process, leading to the collapse of the gel network structure. The surimi gel with 30% ice added showed a uniform, ordered, honeycomb-like three-dimensional network structure. The honeycomb-like gel network structure was easier to lock water molecules, effectively trapped the water molecules in the gel system, and then presented ideal mechanical properties [36]. With the continuous increase in the amount of ice added, the three-dimensional network of gel was gradually replaced by coarse aggregates with large and disordered pores, which might be due to the different unfolding and aggregation speeds of protein. The key factor in the formation of the surimi gel network structure was the relative speed of protein unfolding and aggregation. When the aggregation speed of protein was faster than its unfolding speed, the surimi gel formed a large and rough gel structure [37,38], eventually rendering weak gel properties.

### 3.10. Correlation Analysis

TB tool and SPSS software were used to analyze the correlation between gel properties, water migration, and other indicators of surimi gel with different ice additions. Using TB tool software, 24 indicators at 5 levels were used as variables, and the clustering heat map was obtained by normalizing the variables (Figure 6). The colors were assigned according to the 5 normalized values of each variable. The maximum value was classified as 1, and the value was marked as dark red, and the minimum value was classified as −1, which was marked as dark blue. As the squared Euclidean distance increased, a branch in the red box in Figure 6 was divided into one category, indicating that there was a high correlation among the gel strength, β-sheet, hydrogen bonds, WHC, hydrophobic interactions, disulfide bonds, SH group, breaking force, and P_21_ of surimi gel. The results in Table 4 also confirmed the results of the clustering heat map. The correlation coefficients with the gel strength higher than 0.8 were the breaking force, WHC, β-sheet, hydrogen bonds, disulfide bonds, SH group, and P_21_ of surimi gel, showing that these indicators had a high positive correlation with the gel strength. In addition, the indicators that had a moderately positive correlation (correlation coefficient > 0.5) with the gel strength were the hydrophobic interactions, S_0_-ANS, and actin band density of surimi gel, while T_21_, P_23_, and MHC band density were moderately negative correlation with the gel strength. T_21_, P_23_, and MHC band density represented the binding force of bound water, the peak proportion of free water, and the crosslinking of MHC, respectively. The lower the values of these indicators, the greater the contribution to the gel strength, so these three indexes were considered to have a moderate correlation with the gel strength. It was worth noting that the α-Helix, β-corner, random coil, and hydrophobic interactions of surimi gel all had a high degree of negative correlation with the gel strength, and had significant correlations at *p* < 0.05 or *p* < 0.01, respectively. These indicated that the changes in the protein secondary structure, especially the unwinding of the α-helix, had a great influence on the gel properties of the surimi gel under different amount of ice added.

## 4. Conclusions

The gel strength and WHC of the *N. virgatus* surimi gel with 30% ice added (78.6% moisture content) were significantly higher than those of other treatment groups (*p* < 0.05). The 30% ice addition contributed to the unfolding of the α-helix and the increase of the β-sheet proportion of the protein of surimi gel, resulting in the exposure of the SH group. These promoted the binding of proteins through disulfide bonds and hydrophobic-hydrophobic interactions, forming an ordered honeycomb three-dimensional network structure. In addition, the increases in hydrogen bonds promoted the protein-water interaction, resulting in an increase in the amount and binding force of bound water in the gel, reducing the migration of immobile water to free water, resulting in better gel strength and WHC. When the amount of ice added was insufficient, the surimi gel structure lacked the support of immobile water, resulting in deterioration of the mechanical properties of the gel. However, excessive addition of ice caused faster protein aggregation than unfolding, which was not conducive to the combination of protein-protein and protein-water, forming a large and rough gel structure, resulting in the migration of immobile water to free water and ultimately exhibited weak gel properties.

These findings were consistent with our hypothesis. Therefore, water migration also played a key role in improving the surimi gel properties. Furthermore, the thermal diffusivity of surimi was mainly determined by moisture content (added ice amount) rather than fish species. Therefore, it is speculated that the conclusions of this study were applicable to surimi made from other species of fish.

## Figures and Tables

**Figure 1 foods-10-01815-f001:**
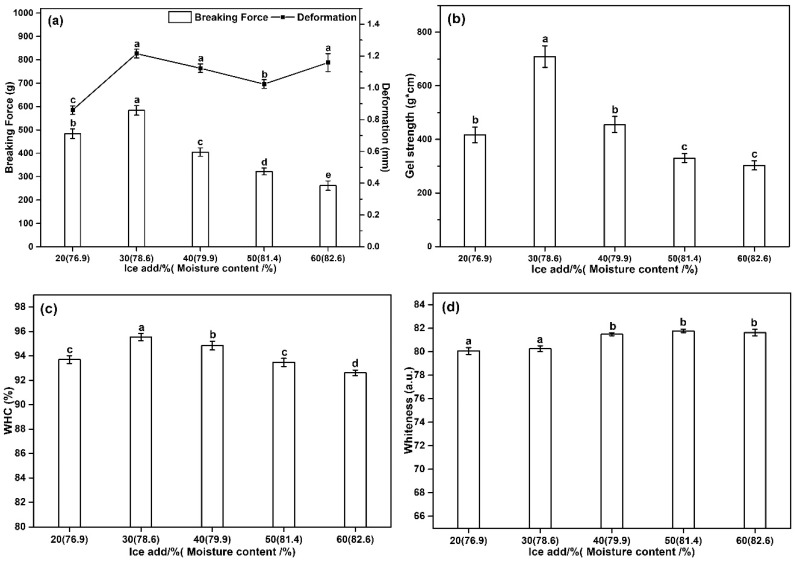
Breaking force and deformation (**a**), gel strength (**b**), WHC (**c**), and whiteness (**d**) of surimi gel with various amounts of ice added. Different letters on the bar at the same indicator indicate significant differences (*p* < 0.05). Bars represent the standard deviation (*n* = 7).

**Figure 2 foods-10-01815-f002:**
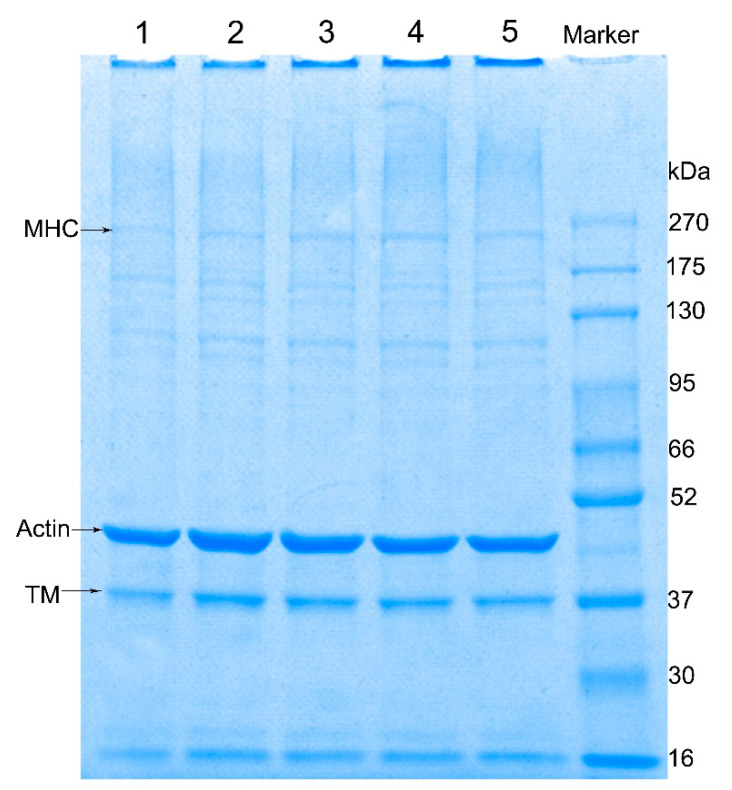
SDS-PAGE pattern of surimi gel with various moisture content. Lanes 1–5 represents the ice add (moisture content) of 20% (76.9%), 30% (78.6%), 40% (79.9%), 50% (81.4%), and 60% (82.6%). MHC: myosin heavy chain; TM: tropomyosin.

**Figure 3 foods-10-01815-f003:**
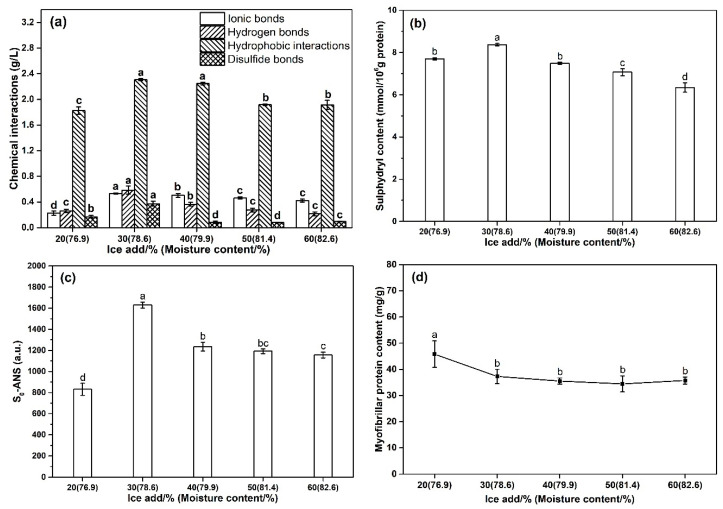
Chemical interactions (**a**), R-SH group content (**b**), S_0_-ANS (**c**), and myofibrillar protein content (**d**) of surimi gel with various amounts of ice added. Different letters on the bar at the same indicator indicate significant differences (*p* < 0.05). Bars represent the standard deviation (*n* = 5).

**Figure 4 foods-10-01815-f004:**
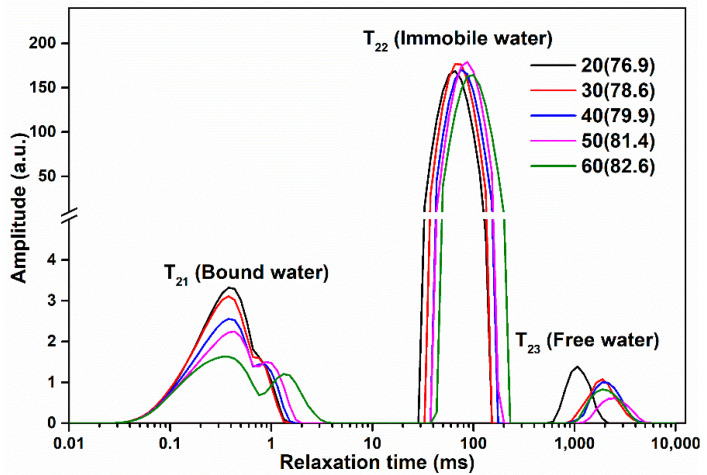
Effects of various amounts of ice added on the distribution of the T_2_ relaxation times of surimi gels.

**Figure 5 foods-10-01815-f005:**
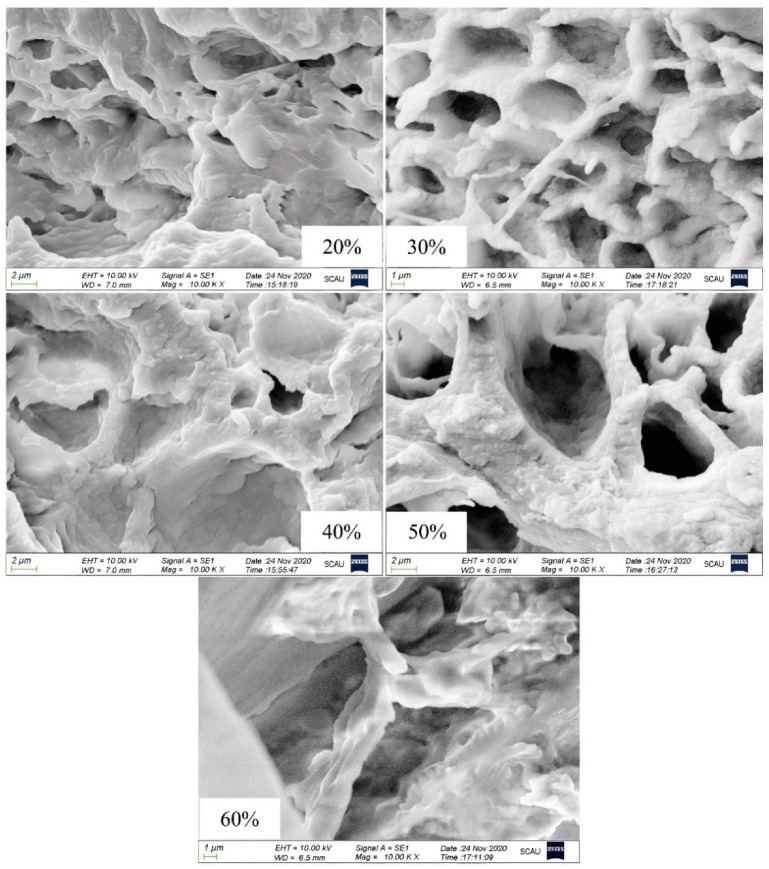
Scanning electron micrograph of surimi gels under various amounts of ice added.EHT: extra high tension; WD: working distance; Mag: magnification.

**Figure 6 foods-10-01815-f006:**
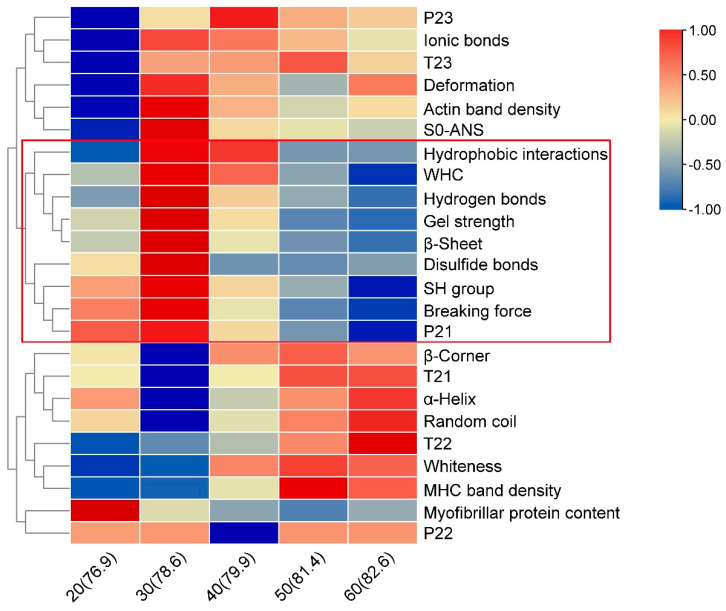
The heat map clustering of surimi gels under various amounts of ice added.

**Table 1 foods-10-01815-t001:** Color, MHC band density, and actin band density of surimi gel with various amounts of ice added.

Ice Adds/% (Moisture Content/%)	*L**	*a**	*b**	MHC Band Density	Actin Band Density
20 (76.9)	82.38 ± 0.74 ^b^	−0.72 ± 0.16 ^a^	9.13 ± 0.50 ^a^	309.39 ± 4.24 ^e^	6104.26 ± 14.73 ^e^
30 (78.6)	81.95 ± 0.50 ^b^	−0.98 ± 0.17 ^b^	8.18 ± 0.18 ^b^	386.51 ± 14.14 ^d^	8241.00 ± 20.51 ^a^
40 (79.9)	83.25 ± 0.18 ^a^	−1.07 ± 0.12 ^b^	7.80 ± 0.18 ^bc^	636.16 ± 7.07 ^c^	7421.15 ± 7.98 ^b^
50 (81.4)	83.47 ± 0.25 ^a^	−1.03 ± 0.05 ^b^	7.65 ± 0.10 ^c^	773.47 ± 2.46 ^a^	7354.86 ± 28.79 ^c^
60 (82.6)	83.08 ± 0.52 ^a^	−1.12 ± 0.10 ^b^	7.22 ± 0.50 ^d^	751.13 ± 5.36 ^b^	7255.34 ± 9.60 ^d^

*L**: lightness; *a**: red-green value; *b**: yellow-bule value. The different superscripted letters of the same column of values indicate significant differences (*p* < 0.05), and the values are expressed as mean ± SD.

**Table 2 foods-10-01815-t002:** Percentage of secondary structures of surimi gels with various amounts of ice added.

Ice Adds/% (Moisture Content/%)	α-Helix (%)	β-Sheet (%)	β-Corner (%)	Random Coil (%)
20 (76.9)	19.77 ± 0.22 ^a^	19.31 ± 0.33 ^b^	42.79 ± 0.08 ^ab^	18.13 ± 0.03 ^a^
30 (78.6)	19.17 ± 0.15 ^b^	20.74 ± 0.32 ^a^	42.37 ± 0.09 ^b^	17.73 ± 0.09 ^b^
40 (79.9)	19.59 ± 0.18 ^ab^	19.44 ± 0.73 ^b^	42.90 ± 0.44 ^ab^	18.08 ± 0.15 ^a^
50 (81.4)	19.79 ± 0.29 ^a^	19.02 ± 0.72 ^b^	42.96 ± 0.22 ^a^	18.23 ± 0.22 ^a^
60 (82.6)	19.93 ± 0.16 ^a^	18.85 ± 0.36 ^b^	42.89 ± 0.14 ^ab^	18.34 ± 0.10 ^a^

The different superscripted letters of the same column of values indicate significant differences (*p* < 0.05), and the values are expressed as mean ± SD.

**Table 3 foods-10-01815-t003:** Water mobility and distribution of surimi gels with various amounts of ice added.

Ice Adds/% (Moisture Content/%)	20 (76.9)	30 (78.6)	40 (79.9)	50 (81.4)	60 (82.6)
T_21_ (ms)	0.40 ± 0.03 ^ab^	0.38 ± 0.00 ^b^	0.40 ± 0.03 ^ab^	0.41 ± 0.03 ^a^	0.41 ± 0.03 ^a^
	-	0.75 ± 0.00 ^b^	0.80 ± 0.06 ^b^	0.92 ± 0.07 ^b^	1.45 ± 0.19 ^a^
T_22_ (ms)	65.79 ± 0.00 ^e^	70.72 ± 0.00 ^d^	75.65 ± 0.00 ^c^	86.98 ± 0.00 ^b^	100 ± 0.00 ^a^
T_23_ (ms)	1158.57 ± 172.60 ^b^	2168.44 ± 301.87 ^a^	2180.91 ± 398.46 ^a^	2357.68 ± 307.66 ^a^	2050.59 ± 267.59 ^a^
P_21_ (%)	3.09 ± 0.07 ^ab^	3.20 ± 0.22 ^a^	2.88 ± 0.17 ^b^	2.66 ± 0.09 ^c^	2.41 ± 0.24 ^d^
P_22_ (%)	96.24 ± 0.23 ^b^	96.67 ± 0.19 ^a^	96.51 ± 0.26 ^ab^	96.86 ± 0.06 ^a^	96.87 ± 0.14 ^a^
P_23_ (%)	0.49 ± 0.01 ^ab^	0.38 ± 0.10 ^b^	0.57 ± 0.10 ^a^	0.43 ± 0.02 ^ab^	0.40 ± 0.08 ^b^

The different superscripted letters of the same row of values indicate significant differences (*p* < 0.05), and the values are expressed as mean ± SD.

**Table 4 foods-10-01815-t004:** Correlation analysis of gel strength of surimi gel with various ice additions and other indicators.

Index	Correlation Coefficient	Index	Correlation Coefficient
Breaking force	0.908 *	SH group	0.900 *
Deformation	0.418	S_0_-ANS	0.717
WHC	0.912 *	T_21_	−0.994 **
Whiteness	−0.650	T_22_	−0.647
α-Helix	−0.983 **	T_23_	0.071
β-Sheet	0.996 **	P_21_	0.824
β-Corner	−0.936 *	P_22_	−0.047
Random coil	−0.994 **	P_23_	0.021
Ionic bands	0.402	MHC band density	−0.725
Hydrogen bonds	0.974 **	Actin band density	0.602
Hydrophobic interactions	0.784	Myofibrillar protein content	0.097
Disulfide bonds	0.920 *		

A correlation coefficient greater than 0.8 indicated that the index was highly correlated with gel strength; Correlation coefficients of 0.5–0.8, 0.3–0.5, and less than 0.3 indicate that the relationship between the index and gel strength was moderate correlation, low correlation, and irrelevance. * and ** indicate that the index was significantly correlated with the gel strength at the level of 0.05 and 0.01, respectively.

## Data Availability

Not applicable.

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
