# Peer review of "Insight into the Effect of Ice Addition on the Gel Properties of Nemipterus virgatus Surimi Gel Combined with Water Migration"

_foods, 2021, doi:10.3390/foods10081815_

Round 1

Reviewer 1 Report

 In this manuscript entitled "Insight into the effect of ice addition on the gel properties of Nemipterus virgatus surimi gel combined with water migration", the authors explored the effect of different amounts of ice added (20-60%) on the gel properties of Nemipterus virgatus surimi gel under the same heat treatment conditions (two-stage heat treatment) and combined with water migration.  The authors suggested the changes in the protein secondary structure, especially the unwinding of the a-helix, had a great influence on the Nemipterus virgatus surimi gel properties under different amount of ice added.  This result gives interesting and valuable information for the research field of surimi gel.  In addition, the authors' conclusions are very significant because of the multifaceted evaluation.  However, there are some problems and flaws in presentation.  I hope that my comments are very useful for the improvement of this research.

Comments

  1. L29-37: These are the sentences of the instruction for author. Please delete.
  2. 2.2. Sample preparation: The authors should indicate how to prepare the ice. In addition, is the temperature in each uniform paste the same after the chopped?
  3. Table 1: The band intensities of MHC and actin bands should be indicated by the intensity of MHC and actin corrected by the intensity of all bands.  In this case, the unit is % (MHC band intensity / total band intensity).
  4. Fig. 1d: The units of the y-axis should be indicated. If there is no unit, it should be arbitrary unit.
  5. Fig. 2: The MLC and M written in the caption is not shown in the Fig. 2.
  6. Table 2: Please review the significant figures for the numbers shown in the Table 2. I feel that the significant figures are high.
  7. Fig. 3c: The units of the y-axis should be indicated. If there is no unit, it should be arbitrary unit.
  8. Fig. 4: Does a.u. mean arbitrary unit? Please explain the abbreviation in the figure caption.
  9. Fig. 6: What are the numbers you are using for heatmap? I think this needs to be explained.
  10. Discussion: In this study, the authors used Nemipterus virgatus as the study material, but can the same conclusion be drawn for other fish species? I think it needs to be considered.

Reviewer 2 Report

General remarks:

This article is very well written and follows the instructions for authors.

The results and conclusions presented in this work make a good contribution for the study of surimi production from Nemipterus virgatus.

My review is just going to point out some general issues with the manuscript.

Revision comments:

Abstract:

- The text presented in the Abstract does not reflect the complexity and diversity of the methods used in this work.

Introduction:

- Authors forgot to delete the first paragraph of the Introduction, which is just the instructions for authors, and not something related with the manuscript.

- The Introduction lacks important information that is relevant for the readers understanding the technicalities of the work, and to frame the socio-economic importance this work.

What is surimi?

Economic importance of Nemipterus virgatus as a fisheries resource, and why is suitable for surimi?

The importance of surimi in the industry of seafood products.

Materials and Methods:

The equation of Whitness (line 111) lacks a reference.

Results:

The chapter “3. Results” should be renamed “3. Results and discussion”, because in this chapter the authors not only present their results, but also discussed them using bibliography.
